# Treatment Algorithm for Cancerous Wounds: A Systematic Review

**DOI:** 10.3390/cancers14051203

**Published:** 2022-02-25

**Authors:** Andrea Furka, Csaba Simkó, László Kostyál, Imre Szabó, Anikó Valikovics, Gábor Fekete, Ilona Tornyi, Endre Oross, János Révész

**Affiliations:** 1Centre of Clinical Oncology and Radiotherapy, Borsod-Abaúj-Zemplén County Hospital and University Teaching Hospital, 3526 Miskolc, Hungary; iszabodr.onkologia@bazmkorhaz.hu (I.S.); avalikovics3@gmail.com (A.V.); feketeg.onkorad@bazmkorhaz.hu (G.F.); oross.onkorad@bazmkorhaz.hu (E.O.); dr.revesz.janos@gmail.com (J.R.); 2Department of Clinical Radiology, Faculty of Health Care, Institute of Practical Methodology and Diagnostics, University of Miskolc, 3526 Miskolc, Hungary; kostyalfed@gmail.com; 3Erzsébet Hospice, Borsod-Abaúj-Zemplén County Hospital and University Teaching Hospital Miskolc, 3526 Miskolc, Hungary; simkocsa@gmail.com; 4Department of Diagnostic Imaging, Borsod-Abaúj-Zemplén County Hospital and University Teaching Hospital Miskolc, 3526 Miskolc, Hungary; 5Department of Human Genetics, University of Debrecen, 4032 Debrecen, Hungary; tornyi.ilona@med.unideb.hu; 6Biosystems Immunolab Zrt., 4032 Debrecen, Hungary

**Keywords:** cancerous wound, treatment algorithm, multimodal aspects, surgery, oncology, radiotherapy, palliative care

## Abstract

**Simple Summary:**

Cancerous wound is a very distressing condition developing mainly in the end-of-life oncological treatment. There is no comprehensive treatment algorithm.

**Abstract:**

Background: In advanced cancer stage the incidence of cancerous wounds is about 5%, and the estimated life expectancy is not more than 6 to 12 months. Without interdisciplinary and individualized treatment strategy, symptoms progress, and adversely influence quality of life. Methods: Authors collected different treatment algorithms for cancerous wound published by wide scale of medical expertise, and summarized surgical, oncological, radiation oncological, nursing and palliative care aspects based on radiological information. Results: Interdisciplinary approach with continuous consultation between various specialists can solve or ease the hopeless cases. Conclusions: This distressing condition needs a comprehensive treatment solution to alleviate severe symptoms. Non-healing fungating wounds without effective therapy are severe socio-economic burden for all participants, including patients, caregivers, and health services. In this paper authors collected recommendations for further guideline that is essential in the near future.

## 1. Introduction

Cancerous wound can be defined as non-healing compound chronic painful wound that arises from cancers and due to increased necrosis and infection, the quantity of malodorous discharge is highly increased [1]. In advanced cancer stage malignant fungating wounds may occur up to 5%, but probably this amount is underestimated since there is no population-based cancer register following the incidence of this condition. The estimated life expectancy is about 6 to 12 months [2,3,4]. This is a complicated condition causing many different distressing symptoms. In most cases discharge from the non-healing wound with malodor, bleeding and surrounding inflammation accompanying severe pain and social isolation leads to a hopeless situation with limited treatment options and predicts very poor prognosis.

Cancerous wounds arise from primary skin tumors, or from metastatic lesions of primary tumors, in most cases from breast, lung, head and neck, and genital malignancies [2,3,4,5]. In this publication Authors collected suggestions from various fields of medicine and analyzed their overlap and made principal conclusions implementing different treatment modalities to the holistic palliative care in order to establish basic guideline for medical practice.

## 2. Materials and Methods

Authors performed a systematic review to identify relevant publications related to treatment options of malignant fungating wound using National Center for Biotechnology Information (NCBI) PubMed database up to 2021. All searches were evaluated according to the title/abstract, then eligible publications were listed in full text format and retrieved. The following keywords were used: “malignant”, “wounds”, “fungating”, “cancerous”, “palliative care”, “radiotherapy”, “surgery”, “and their possible rational combinations.

The inclusion criteria for the search were defined as Original research studies, Clinical Trial, Meta-Analysis, Randomized Controlled Trial, Review, and Systematic Review. The articles were published in English language and all of them were peer-reviewed in a non-predatory medical journal. The collected data introduced adult patients (over 18 years) oncological clinical outcome. Pediatric cases and non-human experimental results, from the view of clinical practice suggestions, were excluded.

We summarized our searching workflow in a PRISMA 2020 Flow diagram on Figure 1 [6].

## 3. Results

Eligible criteria were found in 181 articles, but only 31 articles were published in the last five years. We decided to use these articles as a backbone of our review, and we completed it to acquire a modern multidisciplinary view of cancerous wounds treatment (Figure 2).

### 3.1. Grading of Cancerous Wound, Measurement Tools

Measurement or grading systems help us to estimate and quantify the burden of a certain symptom. Here we list the most useful validated and precisely reproducible scoring systems used in oncology that can be also used for interpretation of malignant fungating wounds.

Latest version of the grading system of the National Cancers Institute’s Common Terminology Criteria for Adverse Events (CTCAE) is useful to make comparable different adverse events [7]. The symptoms possible engaged with malignant fungating wounds that are listed in skin and subcutaneous disorder chapter and their maximum grade: body odor—2, bullous dermatitis—5, dry skin—3, eczema—3, erythroderma—5, fat atrophy—3, pain of skin—3, pruritus—3, skin atrophy—3, skin induration—5, skin ulceration—5, Stevens-Johnson syndrome—5, toxic epidermal necrolysis—5. If CTCAE grade is five it means a lethal complication, therefore malignant wounds may lead to death.

Patient reported outcomes (PROs) have to be integrated into clinicians’ daily practice to evaluate personal distresses and manage symptoms properly. PROs and CTCAE have certain association that was investigated in the literature in details and these scoring systems have metric concordance [7].

PALCARE is a systematic approach to evaluate Prognosis, Advance care planning, Living situation, Comprehensive history, Assessment, Recommendation, Education [2].

Photo documentation with the same device, e.g., mobile phone has to be a standard initial state record. In certain time period, biweekly it must be repeated in the same condition. “Medical green” background is suggested because it provides the best color contrast. Always use millimeter ruler to measure the maximal perpendicular diameters of the wound. We propose, according to RECIST system evaluation, 20% of dimensional change indicates progression or regression. The depth of the wound in case of ulcerating wounds or the maximum height of exophytic wounds has to be calculated to acquire the 3D parameter of the wound. Medical written administration is obligatory, that can be completed with additional standardized photos.

The number of daily wound dress changes, the diameter of wound dressing are also numerical factor that can be comparable [2,5].

The numerical measurement of odor is applicable by TELER system: Treatment Evaluation by A Le Roux’s Method [2]. It ranges between 0–5 points.

As malignant fungating wounds mainly occur in the end-of-life treatment, the cancer outcome has to be calculated, too. For this measurement the modified Glasgow Prognostic Score (mGPS) is very useful, which takes account the C-reactive protein level (less or more than 10 mg/L) and the albumin level (less or more 35 g/L). mGPS 0 means a good prognosis, while mGPS 2 equals with a poor outcome [3].

### 3.2. Radiological Aspects

#### 3.2.1. Radiological Examinations

X-ray can be informative in case of cancerous wound if there is suspicion of fistula formation or perforation. Free air is visualized on X-ray in such condition. X-ray is also diagnostic to evaluate bone involvement and consecutive osteomyelitis.Ultrasonography is useful to detect free air, free fluid. With its help the point for drainage can be marked.CT scan, MRI are sophisticated radiological methods. Use them in the end-of-life treatment only if the result changes the therapeutic decision and prolongs a meaningful life.

#### 3.2.2. Interventional Radiology Can Be Useful in Case of Severe Bleeding

If the nourishing artery is visualized and reachable for radiologist, endovascular embolization is a possible solution.

### 3.3. Surgical Aspects

#### 3.3.1. Surgical Approach Is Not the First Chosen Method to Treat Malignant Fungating Wounds, since It Has Limited Role in Managing Cancerous Wounds

Tumor microenvironments consist of inflammatory cytokines that negatively influence the wound healing. Therefore, surgical removal of malignant wounds impact wound healing failure. An unnecessary surgical removal may enhance tumor progression [8].

Furthermore, moisture associated skin damage and pruritus worsen wound healing [3].

Chrisman et al. declared that is important to recognize, that successful surgical interventions are unrealistic expectations or even burdensome for cancerous wound improvement [9].

Biopsy has to be taken in case on unknown histology or when progression develops during systemic treatment and the new biopsy data could change the therapeutic algorithm, i.e., new mutation with therapeutical consequence.Excision is rare in case of cancerous wound. When such wounds appear and infiltrating the skin the tumorous condition requires systemic treatment, furthermore R0 resection is highly unrealistic. In certain case tumor size may allow for performance in toto excision.Urgent surgery in rare cases would be inevitable. Acute surgical indications are bleeding, ileus (palliative deviation needed), septic condition (requires even amputation) [10], myelon compression, etc. The mortality rate is very high.Negative Pressure Wound Therapy (−100 to −125 mmHg pressure continuously for 24 h) is not routinely advised in malignant fungating wound as it influences the lymphatic drainage and may transfer tumorous cells into the circulation [5]. From the other hand, it may alleviate procedure decreasing symptoms and improving quality of life in end term phase of the disease, therefore in selected cases can be used in palliative care [5,11,12].

#### 3.3.2. Wound Dressing

The most important component of cancerous wound care is to select the appropriate wound dressing, since many aspects can influence our decision.Type of dressing.Antimicrobial effect of dressing is useful due to accompanying infection. Antimicrobial properties of the *silver* dressings are very effective to reduce malignant fungating wound discharge and malodor, respectively [5,13,14]. However, it is to be considered, that radiotherapy is not compatible with silver contain bandages as it influences the ionizing radiation, mainly enhances unwanted side effects. Other antimicrobial bandages were tested in randomized studies to find effective agents. Honey coated foam dressing has the same effect as silver containing bandage [5,14,15]. However, authors do not recommend honey as it cannot be standardized treatment agent.Hydrocolloid foam bandages can ease the pain and help the probable healing circumstances with its moisturizing effect meanwhile it absorbs massive exudation as well.Calcium alginate compress can minimize the bleeding [16].Time of dressing.Proper time for wound dressing change must be implemented the patient’s daily routine together with bathing or hygienic procedures. In case of sever condition, pain prevention is obligate. It is recommended to minimize the frequency of dressing change according to the wound quality.Topicals for hemostasis and antibiotics develop good effect in reduction of malodor and discharge:Topical metronidazol (0.8%) has a very good antimicrobial effect [5,17].Polyhexamethylene biguanide (PHMB 0.2%) has the same result as metronidazole. Both significantly reduce wound odor by day eight [17].6% miltefostine solution is less useful in malignant wounds. It was applied in breast cancer patients with the aim of slowing progression [18].Arsenic trioxide with hemostyptic effect is an easily applicable and cheap agent for topical use [19].Green tea extract as an essential traditional phytotherapy has antimicrobial effect and also suppresses malodor [19,20].Oxymetazoline, an alpha adrenoreceptor agonist has vasoconstrictor sympathomimetic and decongestant effect [21].Etamsylate is a sulfonic acid derivative with vascular protective and hemostatic effect can be directly applied into the wound.Charcoal as absorbent blocks malodor [5].

#### 3.3.3. Bleeding and Hemostasis

It is always important to evaluate the possible reasons of bleeding to prevent sever conditions. The most frequent reasons are: direct vessel invasion, bioburden during dressing removal due to friable tissues, paraneoplastic or chemotherapy induced thrombocytopenia, aplastic condition due to bone marrow infiltration or irradiation, coagulopathy associated with liver involvement, disseminated intravascular coagulation due to any reason, not proper anticoagulation therapy, etc. [3]. Proper hemostasis is challenging, local methods include special dressing, radiotherapy and systemic treatment involves etamsylate, which increases the endothelial resistance of capillaries and promotes platelet adhesion. However, tranexamic acid, a synthetic amino acid derivative of lysine, with antifibrinolytic effect is risky to apply for cancer patients who have thrombogenic paraneoplastic syndrome.

It is also essential to protect surrounding skin area to prevent additional pain.

### 3.4. Radiation Oncological Aspects

#### 3.4.1. Target Volume Delineation

The gross tumor volume (GTV) is the visible macroscopic tumor mass plus adjacent tissues that seem to be involved on topometric CT. The edematous and hyperemic periwound tissues have to be marked by radiopaque agent (wire, gel). This can be the potential microscopically involved area defined as clinical target volume (CTV). In palliative cases CTV not always includes the locoregional lymph nodes, depending on the general condition of patient (accounting risk of deep vein thrombosis, extreme large volume according to whole body surface) and the aim of palliation. Planning target volume (PTV) accounts with relatively wider safety margin due to probable daily set up margin discordance and inter- and intrafractional deviations.

#### 3.4.2. Dose Description

In palliative cases the classical 10 times daily 3 Gy fractionation schedule during weekdays is the most appropriate dose. In special situation 5 × 4 Gy or even 20 × 2 Gy dose can be delivered. Determining the dose, it is always necessary to count with previous in-field radiotherapy. Acceptable the three-dimensional dose volume constraints for organ at risk (OAR) calculated with only rationally short-term Normal Tissue Complication Probability (NTCP) according to the predictive model of Quantitative Analysis of Normal Tissue Effects in the Clinic (QUANTEC) data [22]. It should be taken consideration that these constrains prescribed for conventionally fractionated metric models. In palliative care hypofractionated doses are widespread set as higher daily fraction doses to achieving early radiobiological effect due to the short life expectancy time limit and enormous miserable suffer. With optimization system novel techniques must be incorporated into palliative radiation oncology, providing quality care for patients with malignant fungating wounds.

#### 3.4.3. Energy

Generally, teletherapy is recommended with linear accelerator. The depth or height of the wound affects the proper energy. For very superficial wounds even 6–10 MeV electron can be used with bolus, while for deeper penetration 6–10 MV photon is required. In simple cases 3D conformal radiotherapy is the preferred option. In cases when delicate organs at risk located in the involved area intensity modulated radiotherapy should be applied. Continuous consultation is required with medical physicist and dosimetrists.

### 3.5. Systemic Therapy from a Medical Oncologist’s Perspective

#### 3.5.1. Chemotherapy

In metastatic setting with poor physical condition the less the better. Choose single agent to reduce the risk of unwanted distressing side effects. Choose chemotherapy that spare bone marrow function.

#### 3.5.2. Hormonal Therapy

Hormonal therapy is relatively safe and means less burden comparing other cytostatic agents. Easy to serve them, ideal in outpatient care. Keep it in mind that they increase the risk of deep vein thrombosis and cardiac side effects.

#### 3.5.3. Targeted Therapy

If targeted therapy is available that should be the most proper choice in case of poor performance status. Multigene panels show the different targets, and using even artificial intelligence software the best targetable available drug can be offer. Tyrosine kinase inhibitors, mainly epidermal growth factor receptor blockers may enhance skin ulcerations, vascular endothelial growth factor inhibitors have the risk of thromboembolic events and fistula formations, therefore do not use them in case of malignant fungating wounds.

#### 3.5.4. Immunotherapy

Nowadays more and more immunotherapies are available for clinical use. Firstly, they were used in the treatment of advanced melanoma, e.g., CTLA-4 and PD-1 blockers [23]. In case of squamous cell skin cancer treatment with PDL1 blockers seems effective.

### 3.6. Pain Management Aspects

In spite of high importance of pain relief in malignant wounds only few controlled studies were published analyzing the mechanisms and treatment modalities of pain. Approximately 50–75% of breast cancer patients associated with malignant wound suffer from severe pain [24,25,26].

#### 3.6.1. Pain Mechanism

Diverse pathomechanisms are responsible for various types of pain in malignant wounds. A deeper understanding of these pathomechanisms is a key for determining the appropriate analgesic strategy [26]. The inflammatory component is obvious and often associated with tissue hypoxia, especially in cauliflower-like tumors. These two factors predispose to the development of peripheral sensitization. In many cases, the pain caused by inflammation affects the skin around the wound and correlates with the amount of exudate, the degradation of the wound edges, the presence of polyamines (putrescin, cadaverine) and granulation tissue [24]. Neuropathic pain associated with peripheral nerve injury is common and usually intense. Enlarged tumor mass -especially in the breast- might cause a tensile force on the surrounding soft tissues due to their gravitational effect, which can be the reason of classical nociceptive pain. Partly associated with this and partly with pathological posture for other reasons, the presence of tension-induced myofascial components can be expected in certain muscle groups, most often in the neck and shoulder. Clinically, short-term incidental pain is treated as a separate entity, the so-called breakthrough cancer pain [27] that adversely affects the patient’s quality of life. This special excruciate incidental pain occurs mainly during dressing changes.

#### 3.6.2. Analgesia

To relieve the pain of chronic non-malignant wounds, Price and colleagues developed a correct recommendation, the so-called Wound Pain Management Model in 2007 that is still valid [26]. As far as we know, there is no similar, widely accepted recommendation for cancerous wounds, although local guidelines may exist. Determining the proportion and significance of each component in complex pains is crucial and difficult task. The patient may have pain due to other reasons; the pain relief strategy should be also personalized. It is very important to *prevent* and minimize wound dressing associated pain. If possible, remove necrotic tissue, debridement, perform mechanical-, enzymatic- or osmotic cleaning, and, if necessary, use local or systemic antibiotic therapy which may help to reduce peripheral sensitization, thereby alleviate pain. However, cleaning itself can be painful, so careful consideration of the expected benefits/harms is required. Choosing the right bandage is also important for pain relief [5,13,14,15,16,28].

Pharmacological treatment can be local and systemic. When removing the dressing, in addition to saline hydrating, it may be necessary to use local anesthetics, preferably in the form of cream, gel or hydrogel, which is applied for approximately 20 min before dressing. Although lidocaine molecule changes its polarity in an inflammatory (acidic) environment and penetration the cell membrane is more difficult, a Cochrane systematic review found adequate evidence supporting use of a combination of lidocaine and prilocaine (Eutectic Mixture of Local Anesthetics, EMLA) cream for dressing pain [29]. Topically applied non-steroidal anti-inflammatory drugs (ibuprofen foam, diclofenac gel) have also been shown to be effective, although in both cases the statistical efficacy is lower than expected, NNT (number needed to treat) was 6 [29]. Topical ketamine, amitriptyline, morphine, methadone, buprenorphine, aspirin, capsaicin (0.025 to 0.075%), clonidine 0.1%, and menthol have all been described to reduce wound pain, but evidence regarding safety and efficacy are weak or lacking [28]. In systemic pain treatment, due to the presence of inflammatory pathomechanism, the use of non-steroidal anti-inflammatory drug is logical but may slightly enhance bleeding. The inflammatory and myofascial pain components respond poorly to opiates. In the latter case, infiltration of the affected muscles with local anesthetic, gentle massage and gymnastics can help. In case of a neuropathic pathomechanism, the opiate response can be predicted by simple drug tests, however, adjuvant analgesics are most likely required. Since the statistical efficacy of each agent in neuropathic pain is low [30,31], it is advisable to select the first adjuvant so that it can be combined with others [32]. The patient’s age, comorbidities, and medications should take into account when select these agents. A fully detailed discussion of the issue is beyond the scope of this study, but some specific aspects are worth mentioning. The first choice should be tricyclic antidepressants (NNT 3.5). Duloxetine and gabapentinoids are also effective, though to a lesser extent than NNT (6.4 for duloxetine; 7.1 for gabapentin and 7.7 for pregabalin) [30,33]. Many other drugs have been investigated, but the evidence of efficacy is sometimes questionable [30,31]. Drugs with strong serotonergic activity (SNRIs, clomipramine) may slightly increase bleeding. The edematous side effect of gabapentinoids based on vasodilation is well known. Whether this may have a harmful role to increase the amount of wound exudate is not yet clear. Carbamazepine is a strong inductor in the CYP system, interacts significantly with many drugs (NSAIDs, tramadol, fentanyl, oxycodone, methadone) and may increase bleeding. Valproate inhibits the metabolism of most NSAIDs and may slightly increase bleeding. Systemic lidocaine and ketamine have been frequently used in palliative care for the treatment of neuropathic pain in the last decade, but there is no consensus on the route of administration and applied doses. Efficacy data are also conflicting [34,35,36,37,38,39]. Preventing dressing pain fast acting opioids (transmucosal or parenteral fentanyl, sublingual methadone), parenteral or nasal ketamine and nitrous oxide may help [35,40,41,42].Complementary procedures (relaxation, aromatherapy, music therapy, meditation, etc.) can help reducing anticipation of anxiety and pain, thus inhibit central pain processes [43].Interventional treatments: intrathecal drug administration (opioids, local anesthetics, ziconotide, baclofen, clonidin) may be necessary in a small percentage of patients who are unresponsive to conventional therapies [33].Neuroablative procedure (chemical or thermal) may also be a therapeutic modality in refractory cases. In case of thoracic or abdominal wall infiltration, use of an easy-to-perform intercostal neurolytic blockade should be considered. In case of perineal tumors, if the patient has a stool and urinary deviation or a catheter, sacral neurolytic blockade may give significant help [44,45].

#### 3.6.3. Pruritus and Skin Care

Research on wound pruritus in the scope of neurobiological processes, both experimentally and clinically, has focused primarily on non-malignant conditions (trauma, burns, leg ulcers, epidermolysis bullosa, atopic dermatitis, etc.). Maida found malignant wounds in 67 of 472 cancer patients enrolled in palliative care, with itching rated as the 6th most common symptom with a frequency of 6% [46].

Regarding the development and transmission of the itching sensation, the prevailing view over the past 10 years is that a small proportion of the nonmyelinated C fibers population (approximately 5%) specializes in the perception and transmission of pruritogenic stimuli, with a number of mediators on peripheral nerve endings (histamine, B-alanine, somatostatin, IL-31, serotonin, proteases, substance-P, nerve growth factor) [47].

The non-healing nature of malignant wounds may be an important factor in the significant differences in the incidence of pruritus associated with malignant and non-malignant wounds. During detailed analysis of itching, not only an assessment of its localization, intensity, frequency, duration, provoking or relieving factors may be advisable, but also a description of the itch associated feelings (tickling, stinging, stabbing, pinching, burning, etc.) [48,49].

If provocative factors can be identified (type of dressing, amount and quality of wound exudate, local heat under occlusive dressing, stress) reducing them may also reduce the itching sensation.

Moisture-associated skin damage is caused by overhydration and maceration of skin due to heavy discharge accompanying with infections. Furthermore, it disturbs the periwound area, resulting lympho-vascular damage may lead to edema and enhanced hypoxic layer intensifying the consequences of irritation. This results in inflammation, swelling, pruritus that can develop pain [3]. Nonsteroidal anti-inflammatory drugs, such as aspirin can be useful in case of pruritus.

Wound-associated pruritus is generally considered to be histamine unresponsive [48], although some pruritoceptive components are expected to be due to complex pathomechanism. The evidence of antipruritogenic effects of medications used in neuropathic pain is extremely modest, the patient’s other symptoms (sleep disturbance, neuropathic pain components, depression, possibly nausea, co-morbidities, and underlying medications) should be considered to determine the optimal non-oncological systemic therapy. Certain antidepressants (doxepin, amitriptyline, mirtazapine, paroxetine) and anticonvulsive drugs (gabapentinoids, sodium channel blockers, valproate) are advised. There is a good experience with mu receptor antagonists and kappa agonists for the relief of chronic pruritus of other origin, but for neuropathic or for wound associated itch, their use is currently not recommended. Based on the available data, an evidence-based recommendation to alleviate the pruritus of tumor wounds is currently not available and further research is needed.

Relaxation, even with aromatherapy, can also have a positive effect on relieving symptoms [50].

Topical treatments are known to relieve chronic itching, in addition to moisturizers, neutral soaps and natural substances. In a recently published review, Andrade discussed in detail the formulations studied so far in clinical trials (clobetasol, mometasone, menthol, camphor, tacrolimus) [50]. Most of these have been tested for pruritoceptive pruritus. In the case of tumor wounds, mostly periwound application is considered. In case of ulcerative tumors, the use of local anesthetics on the wound surface is also common in palliative practice [50].

### 3.7. Psychosocial Aspect

All cancer patients and their caregivers must be screened by onco-psychologist to explore the need of adjuvant mental help. The two main psychological problems among patients suffering from cancerous wound are isolation and depression. Promoting proper sleep is also crucial.

The psychosocial support is also important for analgesia because it can help to increase the activity of descending analgesic pathways. All of the above-mentioned pain mechanisms are associated with an aggravating role of psychosocial factors. Anxiety in relation to anticipation of impending pain can enhance pain intensity and reduce pain tolerance [43].

## 4. Discussion

The devasting impact of malignant fungating wounds is invaluable [2,3,9]. The compound symptoms and magnitude of suffer need precise overview for medical professionals. Always start to treat the most distressing symptom, then alleviate other factors. Comparison of different malignant wounds is very difficult since they have diverse origin. Scoring and grading systems are useful to make comparable each subject to themselves [2,3,51].

Photo documentation with measurement tools is helpful to follow the tendency of wound development and temporal follow up. In the near future, artificial intelligence will probably help with photo toolbar in decision making. Mobil applications are under development phase with no academic accreditation jet but will be part of palliative care.

There is a lack of consensus to guide clinicians [52]. Scientific evidence of data published in connection with cancerous wound is at very low level due to weak quality of methodology, incomparable cases, and lack of interdisciplinary consultation. Therefore, correct meta-analysis is still missing but there is solid evidence that longer survival time can be reached in patients with reduced wound size [15].

Modern wound dressings are available with moisture techniques using hydrocolloid foam dressing containing silver. Topical agents maybe a useful but combining them can lead ineffectiveness or unwanted side effects due to interaction of their components. Systemic antibiotic treatment is still questionable [53].

Novel radiotherapeutic techniques must be incorporated into palliative radiation oncology, too, providing quality care for patients with cancerous wounds.

In this publication Authors collected reviewed articles and recommendations regarding to palliative care of cancerous wounds and highlighted the most important messages in structured aspects for clinicians and caregivers. This collection could be a certain basis for fundamental protocol and mandatory clinical guideline of cancerous wound care. In the future, inter- and multidisciplinary teamwork is needed to help in personalized medicine in such cases, which is highly challenging and multifaceted project.

## 5. Conclusions

Patients with cancerous wound need comprehensive holistic treatment protocols involving not just palliative care givers but even radiation and medical oncologist together with surgeons. Complex multidisciplinary teamwork in the key issue from the benefit of successful or reasonable therapy [54]. All medical professionals have to know how to alleviate this distressed condition, therefore the guideline composed in this paper can be a proper compass.

## Figures and Tables

**Figure 1 cancers-14-01203-f001:**
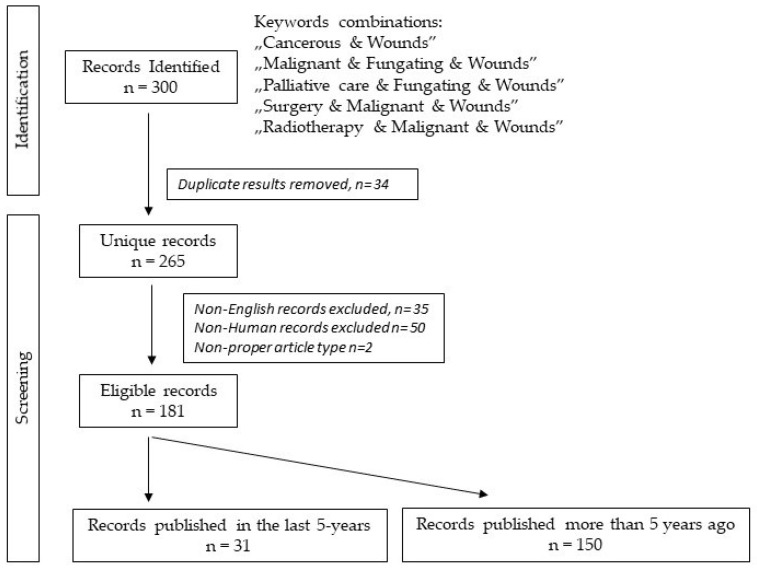
Key terms and searching strategy flow diagram.

**Figure 2 cancers-14-01203-f002:**
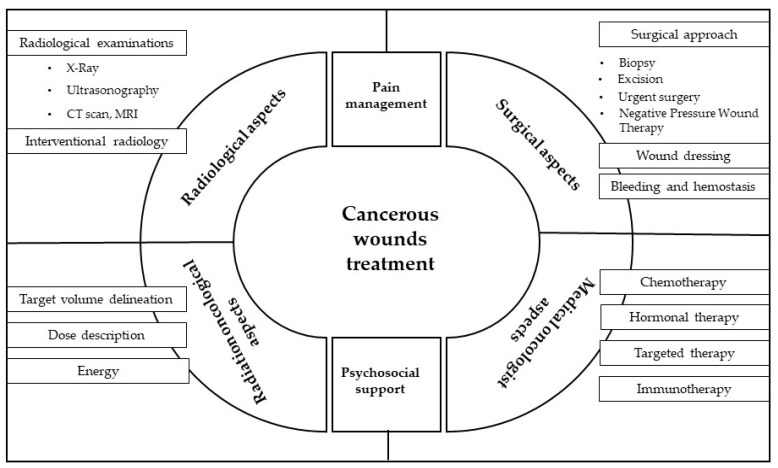
Multidisciplinary view of cancerous wounds treatment.

## Data Availability

The data presented in this study are available on request from the corresponding author.

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
