# Peer review of "Treatment Algorithm for Cancerous Wounds: A Systematic Review"

_cancers, 2022, doi:10.3390/cancers14051203_

Round 1

Reviewer 1 Report

In the current manuscript, Furka et al provided comprehensive review for the treatment regimen on cancerous wounds. The authors had collected different treatment algorithms for cancerous wound published by wide scale of medical expertise, and summarized surgical, oncological, radiation oncological, nursing and palliative care aspects based on radiological information. Such Interdisciplinary approach with continuous consultation between various specialists has shown promise to solve or ease the 25 hopeless cases. This review thus provided the medical professionals with suitable approaches one could use to alleviate this distressed condition.

Reviewer 2 Report

With great interest I have read the manuscript titled "Treatment algorithm for cancerous wounds" submitted by Furka and colleagues.

The authors have produced a very good compilation which addresses all the different aspects in the diagnosis, multimodal treatment and prevention of cancerous wounds and the related complications. This is a very important and crucial topic which still remains a therapeutic challenge for the confronted physician/ surgeon as I know out of my daily surgical routine. However  I have no  majoir queries to report.

Minor queries:

Line 201 : "It is always..." (typo)

Line 212: "It is also essential to protect..." (typo)

Reviewer 3 Report

The authors reviewed the treatment algorithm for cancerous wounds. The review is interesting, however, I have some concerns to solve before publication.

  1. There are no new findings in this review.
    If there are, please highlight them. Also, is this systematic? if so, please add that in the title.
    2. The abstract is POOR. Please make it more complete. Please enrich it, especially including the introduction.
    3. Please include literature on the poor prognosis of poor wound healing after surgery and describe how to deal with it.
    4. It is common knowledge that cancer should be treated with team medicine. Emphasize the topics in this study and contrast their advantages and disadvantages.

Round 2

Reviewer 3 Report

The authors replied to my concerns well, so the manuscript is suitable for publication.